# Digital Media Use in Association with Sensory Taste Preferences in European Children and Adolescents—Results from the I.Family Study

**DOI:** 10.3390/foods10020377

**Published:** 2021-02-09

**Authors:** Elida Sina, Christoph Buck, Wolfgang Ahrens, Stefaan De Henauw, Hannah Jilani, Lauren Lissner, Dénes Molnár, Luis A. Moreno, Valeria Pala, Lucia Reisch, Alfonso Siani, Antonia Solea, Toomas Veidebaum, Antje Hebestreit

**Affiliations:** 1Leibniz Institute for Prevention Research and Epidemiology-BIPS, 28359 Bremen, Germany; sina@leibniz-bips.de (E.S.); buck@leibniz-bips.de (C.B.); ahrens@leibniz-bips.de (W.A.); jilani@leibniz-bips.de (H.J.); 2Institute of Statistics, Faculty of Mathematics and Computer Science, University of Bremen, 28359 Bremen, Germany; 3Department of Public Health and Primary Care, Ghent University, 9000 Ghent, Belgium; stefaan.dehenauw@ugent.be; 4Institute for Public Health and Nursing Research—IPP, University of Bremen, 28359 Bremen, Germany; 5Department of Public Health and Community Medicine, University of Gothenburg, 40530 Gothenburg, Sweden; lauren.lissner@gu.se; 6Department of Pediatrics, Medical School, University of Pécs, 7623 Pécs, Hungary; denes.molnar@aok.pte.hu; 7GENUD (Growth, Exercise, Nutrition and Development) Research Group, Instituto Agroalimentario de Aragón (IA2), Instituto de Investigación Sanitaria Aragón (IIS Aragón), Centro de Investigación Biomédica en Red Fisiopatología de la Obesidad y Nutrición (CIBERObn), University of Zaragoza, 50009 Zaragoza, Spain; lmoreno@unizar.es; 8Department of Preventive and Predictive Medicine, Fondazione IRCCS, Istituto Nazionale dei Tumori, 20133 Milan, Italy; valeria.pala@istitutotumori.mi.it; 9Department of Management, Society and Communication, Copenhagen Business School, 2000 Frederiksberg, Denmark; lre.msc@cbs.dk; 10Institute of Food Sciences, National Research Council, 83100 Avellino, Italy; asiani@isa.cnr.it; 11Research and Education Institute of Child Health, 2035 Strovolos, Cyprus; toniasolea@yahoo.com; 12Department of Chronic Diseases, National Institute for Health Development, 11619 Tallinn, Estonia; toomas.veidebaum@tai.ee

**Keywords:** food preference, internet, smartphone, screen-time, digital marketing, I.Family study, taste preference, children

## Abstract

Digital media (DM) influences children’s food choice. We aim to investigate associations between DM use and taste preferences (TP) for sweet, fatty, bitter, and salty in European children and adolescents. Individuals aged 6–17 years (N = 7094) providing cross-sectional data for DM use: television (TV), computer/game console (PC), smartphone and internet, were included. Children (6 to <12 years) and adolescents (≥12 years) completed a Food and Beverage Preference Questionnaire; scores were calculated for sweet, fatty, salty and bitter preference and categorized (high vs. low). Logistic regression was used to calculate odds ratios as association measures between DM exposure and TP. On average, individuals used media for 2.4 h/day (SD = 1.7). Increasing exposures to DM were associated positively with sweet, fatty and salty TP, while inversely with bitter preference. In female adolescents, DM exposure for >2 h/day was associated with sweet (OR = 1.27, 95% CI = 1.02–1.57) and fatty preference (OR = 1.37; 95% CI = 1.10–1.70). Internet exposure was inversely associated with bitter preference, notably in male adolescents (OR = 0.65, 95% CI = 0.50–0.84), but positively associated with salty preference (OR = 1.29, 95% CI = 1.02–1.64). DM exposure was associated with sweet, fatty, salty and bitter TP in children and adolescents, serving as the basis for future longitudinal studies to shed light on the underlying mechanism by which DM exposure may determine eating habits.

## 1. Introduction

The increasing prevalence of childhood obesity worldwide is mainly driven by modifiable lifestyle risk factors including unhealthy dietary intake [1] and adoption of sedentary behaviors such as use of screen media devices [2]. One of the core recommendations of the World Health Organization (WHO) to halt childhood obesity is to reduce children’s intake of foods high in fat, salt and sugar (HFSS foods) [1]. It is well-documented that food intake is determined by taste preferences (TP) which are established during childhood and adolescence and are meant to track into adulthood [3]. These are influenced by genetic [4] and environmental factors, including diet quality [5], culture [6], and home and non-family-shared environment [7]. Evidence shows that children learn to prefer energy dense foods over energy-diluted versions of the same foods [8]. This behavior may promote adverse health effects in the current obesogenic environments with the omnipresence of HFSS foods, together with a high exposure to these foods in the digital environment.

Remarkably, prolonged use of screen-media devices (i.e., television (TV)) has been described as a significant contributor to poor eating habits in children and adolescents, including higher propensities to consume sweets and fatty foods [9], and reduced intake of fruits and vegetables [10], determining the development of overweight and obesity [11]. TV and video gaming can lead to unfavorable adiposity markers through prolonged bouts of sedentary behavior [12] and increased eating while viewing [13], with media distracting from or obscuring the feelings of satiety [14]. Another mechanism is the persuasive effect of food marketing targeting children increasingly on multiple digital media (DM) channels, such as computers, tablets and smartphones. These channels provide ubiquitous access to internet, social media platforms and advergames [14]. The WHO has identified digital marketing for unhealthy foods as detrimental to children’s and adolescents’ health [15]. Food commercials embedded in animated programs increase immediate eating of advertised food products (e.g., snacks) [16], even in brief 30-s TV commercials [17].

Highly appetizing food pictures and videos in food-related TV programs, advertisements and smartphone screens may stimulate a myriad of neural, physiological and behavioral responses [18]. Viewing pictures of food compared to non-food cues is associated with increased secretion of grehlin, the strongest orexigenic hormone increasing appetite and caloric intake [19], and higher visual attention, the latter shown via an eye-tracking study in children [20]. Furthermore, branding of foods and beverages altered young children’s actual taste perceptions in side-by-side taste tests [21], especially among those watching more television. A recent study observed that eating while watching TV was associated with lower preference for bitter tasting foods and higher preference for sweet tasting foods, suggesting that TV watching could lead to a reduced attention to the sensory characteristics of food [22].

Studies indicate that merely watching TV food commercials compared to non-food ads [23] can activate taste and reward-related brain areas. Using an electro-encephalography (EEG), Ohla et al. (2012) showed that images of calorie-dense foods can enhance hedonic taste evaluation [24]. A hedonically neutral electric taste signal elicited by a small current applied to participants’ tongues was rated as more pleasant after viewing high-calorie food images than after viewing low-calorie, with effects being stronger in the insula and the orbitofrontal cortex, i.e., the reward processing and decision-making brain areas.

The above evidence suggests that digital media and exposure to food images provided through them can modulate taste perceptions and preferences. However, epidemiological studies evaluating how exposure to DM in real-life settings (i.e., outside the lab) influences children’s taste preferences are lacking. We aim at closing this research gap by evaluating associations between different types of DM including TV, computer/game console (PC) and smartphone use, as well as the exposure to internet content and children’s and adolescents’ taste preferences for sweet, fatty, salty, and bitter, in a large sample from 7 European countries.

## 2. Materials and Methods

### 2.1. Study Design and Setting

This cross-sectional study was conducted in the framework of the I.Family study, aimed at investigating determinants of eating behaviors in European children and adolescents and their parents [25]. The I.Family study was conducted in 2013–2014 using standardized instruments and protocols in Belgium, Estonia, Cyprus, Hungary, Italy, Germany, Spain and Sweden, including 7841 participants, aged 2–17 years. A sub-sample of children (7105) aged ≥6 years completed a Food and Beverage Preference Questionnaire across all study centers (excluding Belgium) to measure their preferences for specific food-groups. In order to correct for misreporting bias, children with extremely high DM use (>50 h/week, N = 100) and not using DM at all (or missing, N = 343) were excluded (Appendix A). A total of 7094 children and adolescents were included in the present study. Information on duration of DM use, its specific types and dietary behaviors were obtained for all participants. Questionnaires were developed in English, translated into local languages and back-translated to English to check for errors. Written informed consent was obtained from adolescents and from parents of all children. Children below the age of 12 years were orally informed by field workers before each examination and were asked for their oral assent. Ethical approval was obtained from local institutional review boards at each study center.

### 2.2. Data Collection

#### Core Questionnaire and Assessment of Media Use

Data on age, sex, country of residence and migration background were self-reported by adolescents and proxy-reported by parents of younger children (i.e., children aged <12 years), respectively using the teen and the parental version of the questionnaire, which have been tested for validity and reproducibility [26]. Parents self-reported their highest educational level, based on the International Standard Classification of Education (ISCED) [27] which was classified in three main categories: “low”, “medium” and “high”. Children’s migration background was assessed based on whether one, both or none of their parents were born outside of the respective country of residence.

Participants reported their time spent with different media types, including TV/DVD/video, computer/game consoles (PC) and use of internet on weekdays and on weekend days as: not at all, less than 30 min/day, 30 min to 1 h/day, about 1–2 h/day, about 2–3 h/day and >3 h/day” in line with the methodology used in previous studies [28]. Internet users could also choose the option of “I’m online more or less all day/night”. For PC use, we explicitly asked “How long do you usually sit at a computer/game console per day? (Please disregard the time spent on internet-use.)”, in order to obtain precise information regarding the passive use of PC and game consoles, thus preventing potential overlap with internet use. Assessment of media use did not distinguish between the time used with specific media for recreational and/or educational purposes.

Total digital media (DM) use was calculated as the weighted average of the durations reported for weekdays and for weekend days, expressed in total minutes/week and converted into total hours/day. For the present analyses, the daily duration of DM exposure was categorized as: ≤1 h/day, 1 to ≤2 h/day, 2 to ≤3 h/day and >3 h/day to assess trends of media exposure and to better reflect the original variable. Similarly, daily duration use of single media types was classified, hereinafter referred as TV viewing, PC use and internet exposure. Furthermore, using the question: “Thinking only about yesterday, about how much time did you spend watching TV shows, movies or music videos on a cellphone?”, children were asked to recall the time spent with cellphones (hereinafter smartphone use). On a 5-point Likert-scale, answers ranged from 0 meaning “not at all” to 5 meaning “more than 3 h/day”. Smartphone use was categorized similarly to the other media types.

### 2.3. Assessment of Sensory Taste Preferences

Children and adolescents (6–17 years) completed a Food and Beverage Preference Questionnaire aiming at assessing preferences for sweet, fatty, salty and bitter taste based on a list of selected food items and beverages [29]. Sour-tasting foods were not included, as we aimed to evaluate sensory taste preferences that are linked to the current obesogenic diets, characterized by foods low in fiber [30] and high in fat, sugar and salt content (HFSS foods) [31]. Hence, preferences for sweet, fatty and salty foods were measured as a proxy for unhealthy food preferences [32] and bitter preference as a proxy for healthy food preferences [33]. To ensure the availability of food items in all countries, a pre-test was conducted [34]. Photographs of 63 various food items considered appropriate for all age groups were included in the final questionnaire: single foods (e.g., spinach, banana, broccoli), condiments (e.g., mayonnaise, nougat spread), mixed foods (e.g., sausage, kebab) and drinks (e.g., lemonade). Participants indicated how much they like the taste of the foods/drinks in the photographs, using a 5 point-Likert scale, from 1 meaning “Do not like at all” to 5 meaning “I like very much”. Children who had never tried (or did not know) a specific type of food indicated the respective option.

### 2.4. Taste Preference Scores

A sex- and age-specific factor analysis was conducted to assign specific foods and beverages to the respective taste modalities: sweet, fatty, bitter and salty, and to account for the factorial structure of food preference. Foods and beverages that were recognized/tasted by less than 75% of participants were excluded, such as: asparagus, black coffee, Brussels sprouts, grapefruit etc. Further details have been previously described [34]. The TP scores were calculated as the sum of the rating for foods/drinks assigned to each taste category and divided by the total number of food/drink items included in that specific group. Following the age and sex-specific factor analysis, taste preference scores were calculated separately for males and females of two age groups (<12 years, hereafter referred as children, vs. ≥12 years, referred as adolescents), to control for age and sex discrepancies in food preference. The age of 12 years was chosen as the median age for puberty onset, where changes in child’s anatomy and psychological processes occur [35] (e.g., in the gustatory and olfactory system), and environmental factors such as peer pressure might also influence TP [36]. Additionally, children’s ability to distinguish advertisements from other media content starts from the age of 12 years [37]. The four sub-groups’ scores (male children, male adolescents, female children and female adolescents) were merged into one unique score for each taste modality in order to create a non-stratified taste preference score which would be used as the dependent variable in the models assessing the impact of media use on TP in all children and adolescents. Based on within-sample median values (median = 4 for sweet, fatty and salty preference; median = 3 for the bitter taste preference), each of the four TP scores was categorized as “high” vs. “low” preference. The sample size for the bitter TP was slightly lower compared to the other taste modalities, due to missing values, as a lower number of bitter foods were included, i.e., children tend to recognize bitter tasting foods less compared to sweet or salty tasting ones.

### 2.5. Assessment of Dietary Patterns

To assess diet quality, a healthy diet adherence score (HDAS) was developed. Using pa food frequency questionnaire (FFQ), previously tested for relative validity and reproducibility [38,39], participants indicated the frequency of consumption of 59 different foods items, beverages and mixed dishes in a typical week during the preceding four weeks. Answer options varied from ‘never/less than once a week’, ‘1–3 times/week’, ‘4–6 times/week’, ‘1 time/day’, ‘2 times/day’, ‘3 times/day’ to ‘4 or more times/day’. The description of food items was standardized across countries; examples of country-specific foods were included for a certain food item, to account for cultural discrepancies in food intake. The score was calculated for children with ≥50% of non-missing food items. The HDAS was developed as a composite score to reflect the adherence to the healthy dietary guidelines common across all the participating countries including high consumption of fruits and vegetables (at least 400–500 g/day), limited intake of refined sugars and fat (especially saturated fats), consumption of whole meals, and of fish two–three times per week [40], as established by Waijers et al. (2007) [41]. The score ranged from 0 to 50 and was dichotomized based on median value as “high” vs. “low” adherence to assess the broad concept of healthy diet adherence and to better interpret the data. Based on the FFQ, we additionally assessed the frequency of snack food consumption (times/day), calculated from the frequencies assigned to the following food and drink items: “sweetened drinks”, “chocolate or nut based spread”, “crisps, corn crisps, popcorn”, “chocolate, candy bars” and “candies, loose candies, marshmallows”. Based on within-sample median, children’s snack consumption was classified as “high” vs. “low”.

### 2.6. Anthropometric Measurements

Each child was measured for weight and height in the morning, in light clothing and in fasting status. Weight was measured using a Tanita scale (TANITA Europe GmbH, Sindelfingen, Germany) to the nearest 0.1 kg, while height was measured using a portable stadiometer (Seca GmbH & Co. KG., Hamburg, Germany) to the nearest 0.1 cm. Body Mass Index (BMI) was calculated as weight divided by squared height and transformed into age- and sex-specific z-scores for all children and adolescents. Participant’s weight status was categorized according to the cut-offs of Cole et al. (2012) [42] as thin/normal weight vs. overweight/obese.

### 2.7. Sweet and Fat Intake Propensity

The sweet and fat intake propensities were calculated to reflect the proportion of sweet and fatty foods in children’s diets [9]. The sweet intake propensity was calculated as the proportion of consumed foods/drinks with high sugar content by dividing the sum of the weekly frequency of intake of corresponding foods (e.g., jam, nut-based spreads, chocolate, fruit juice, biscuits, as well as items with added sugar: milk, yoghurt, fresh fruits, drinks, cereal products etc.) by the total frequency of all foods/drinks items included in the FFQ and multiplied by 100. This allowed us to avoid a classification bias by misclassifying children in the high-sugar or high-fat groups only because they have a high frequency consumption of all types of food [9]. The score ranged from 0%-100%. A value of 50% for the sweet intake propensity indicates that half of the reported food consumption frequencies included foods rich in sugar content. The fat intake propensity score was similarly calculated, based on the consumption of foods high in fat including whole-fat milk and yoghurt, cheese, butter, mayonnaise, meat products, fried fish, savory snacks, etc. The scores were dichotomized as “high vs. low” intake propensity at the median value (22.5 for sweet intake propensity and 25.7 for fat intake propensity).

### 2.8. Statistical Analyses

The proportion of children meeting the media use guidelines, as recommended by WHO, i.e., ≤2 h/day of media use for children older than 5 years [2,43], was identified. Descriptive analyses were conducted to explore differences in the sample characteristics (in the number (N) and percentage meeting the DM use guidelines) and sex, including age groups (children vs. adolescents), parent’s educational level, weight status (thin/normal weight vs. overweight/obese), country, migration background, diet quality (HDAS), snack consumption, sweet and fat intake propensities and specific taste preferences. Furthermore, differences in duration of single media types used (four categories: ≤1 h/day, 1 to ≤2 h/day, 2 to ≤3 h/day and ≥3 h/day) by age groups and sex were evaluated. To assess the associations of exposure to different durations of DM and its specific types with TP, odds ratios were calculated by logistic regression, adjusting for covariates: age group (children vs. adolescents), sex (males vs. females), parental educational level (low, medium and missing vs. high), country, migration background (one parent, both and missing vs. none of the parents), diet quality (low vs. high HDAS) and snack frequency intake (high vs. low). In a second step, models were further adjusted for weight status, to take into account the role of BMI. The analyses between the single media types and TP were restricted to children and adolescents actually using that specific media on a daily basis, to make better use of the available data (TV viewing: N = 7052, PC use: N = 5738, smartphone use: N = 3572; internet exposure: N = 6007).

### 2.9. Stratified Analyses by Sex and Age Group

To explore the mediating role of sex (males vs. females) and age (children vs. adolescents), the study population was stratified accordingly and associations were examined based on logistic regression across four strata while adjusting for the remaining covariates, including age, as a continuous variable, in order to control for residual confounding within age group strata. Due to the small sample size across different strata, the media use variables were dichotomized based on WHO recommendation for media use in children >5 years old. Consequently, for the stratified analyses only, participants were classified as “high- >2 h/day” vs. “low- ≤2 h/day” media users.

### 2.10. Sensitivity Analyses

Taking into account that children’s propensity to consume high-fat [9] and high-sugar foods [44] is associated with children’s screen habits as well as fatty and sweet taste preference [29], we considered the mediating role of sweet and fat intake propensities in sensitivity analyses. As a first step, we investigated the association of DM exposure durations (in four categories) with sweet and fatty TP by stratifying the whole sample by sweet and fat intake propensity respectively, based on logistic regressions, while adjusting for covariates. In a second step, we additionally stratified by sex and age group, to consider differences between male and female children and adolescents. Yet, due to the small sample size across strata, the media exposure was considered in two categories only (≤2 h/day vs. >2 h/day).

Odds rations (OR) and 95% Confidence Intervals (CI) were calculated and the level of statistical significance was set at α = 0.05. The statistical software SAS, version 9.4 (Statistical Analyses System, SAS Institute Inc., Cary, NC, USA) was used to perform all statistical analyses.

## 3. Results

A total of 7094 children and adolescents were included in the final analyses (50.7% females). The majority (56.6%) were younger than 12 years. Detailed characteristics of the study population are described in Table 1. Overweight/obese children and adolescents made up 27.5% of the analysis population. On average, participants spent more than 2 h daily in front of screens (mean = 2.4; SD = 1.37) with 54.8% of them exceeding the guidelines (respectively, 44.2% of young children and 68.5% of adolescents). The duration of media use increased with age and differences were observed between males and females (Appendix A). A quarter of all children and adolescents watched TV and 7.6% of them used PC for >2 h/day respectively (Appendix A). Two out of ten children and adolescents (19%) were exposed to internet content daily for >2 h. Half of participants used a smartphone (17% of them used it for >2 h/day). Circa 60% of the study sample had high preference for sweet, fatty and bitter taste, while 52% of them had high salty taste preference. Approximately half of participants had low diet quality (HDAS) and high propensities for sweet and fatty foods, while 47.5% had high intake of snacks.

### 3.1. Association between Media Use and Sweet Taste Preference

The adjusted logistic regression analyses showed a positive trend in the association between increasing durations of DM exposure and sweet TP (Table 2). Exposure for >3 h/day to DM was positively associated with increased sweet preference (OR = 1.23; 95% CI = 1.03–1.46). Further adjustment for weight status, did not attenuate the associations between media exposure and sweet TP (results not shown). In the stratified analyses by sex and age groups, the association remained positive in adolescents with high DM exposure (>2 h/day), for both males and females (respectively 25 and 27% higher odds), compared to those with low DM exposure (≤2 h/day) (Table 3). These associations remained after stratification by propensity to consume sweets, indicating that DM use was positively associated with sweet TP in adolescents, both in the high and low sweet intake propensity groups (Appendix A). Prolonged TV viewing was positively associated with sweet TP across all strata, particularly in female children (OR = 1.31; 95% CI = 1.02–1.69). A positive trend was observed in the association between high smartphone use (>2 h/day) and high sweet TP in all participants, particularly in young children (male children: OR = 2.52; 95% CI = 0.98–6.50; female children: OR = 1.43; 95% CI = 0.73–2.79).

### 3.2. Association between Media Use and Fatty Taste Preference

The adjusted regression analyses showed that exposure to DM for durations >1 h/day was associated with fatty TP (Table 2) in all children and adolescents (1–2 h/day: OR = 1.19; 95%CI = 1.01–1.41; >3 h/day: OR = 1.40, 95% CI = 1.18–1.67). Further adjustment for weight status, did not attenuate the associations between DM exposure and fatty TP (results not shown). After stratification by sex and age, the association remained positive both in male and female adolescents. In the sensitivity analyses, after stratification by fat intake propensity, high DM exposure in adolescents was associated with high fatty TP, in the low and high fat intake propensity groups (Appendix A). Watching TV, using a PC and being exposed to internet content for >2 h/day was associated with high fatty TP in all participants, especially in female adolescents (TV: OR = 1.28; 95% CI = 1.02–1.61; PC: OR = 1.83; 95% CI = 1.21–2.76; internet: OR = 1.37; 95% CI = 1.10–1.71 (Table 4)). Smartphone use for >2 h/day was associated with increased fatty TP in all children and adolescents of both sexes.

### 3.3. Association between Media Use and Bitter Taste Preference

Increasing durations of exposure to DM as well as its single types (TV, PC, internet and smartphone) were inversely associated with bitter TP (Table 2), after adjusting for covariates. Exposures of 1–2 h/day to DM and internet in our cross-sectional sample were respectively associated with 17% and 15% lower odds for preferring bitter tasting foods, compared to ≤1 h/day DM use. The odds for bitter TP in all children reduced to 30% for exposures to DM longer than 3 h/day (OR = 0.72, 95% CI = 0.60–0.87). TV viewing for >2 h daily (Table 5), was inversely associated with preference for bitter taste in male children and adolescents, but not in females. Additionally, in adolescent males, negative associations with bitter TP were observed when they used PC (OR = 0.65; 95% CI = 0.48–0.87), smartphone (OR = 0.68, 95% CI = 0.49–0.94) and internet (OR = 0.65; 95% CI = 0.50–0.84) for >2 h/day. The associations between media types and bitter TP did not attenuate after further adjustment for weight status (results not shown).

### 3.4. Association between Media Use and Salty Taste Preference

Exposure of children and adolescents to DM and TV content for longer than 3 h/day (Table 2) showed a positive trend in association with salty TP (respectively: OR = 1.15, 95% CI = 0.96–1.37; OR = 1.19, 95% CI = 0.93–1.52), compared to low DM exposure (≤1 h/day). Further adjustment for weight status, did not attenuate the associations between media exposure and salty TP (results not shown). After stratification by sex and age, associations remained positive in female children only (Table 6). PC and smartphone use for longer than 2 h/day in female children was positively associated with high salty TP. Additionally, we observed positive associations between increasing durations of internet exposure and salty TP in all participants (Table 2) and in adolescent males in particular (OR = 1.29, 95% CI = 1.02–1.64).

## 4. Discussion

To our best knowledge, this is the first epidemiological study investigating the association of media use patterns with sensory taste preferences in children and adolescents. Our results indicated that European children and adolescents spent 2.4 h/day on average in front of screens, with 54.8% of them exceeding the WHO guidelines. Our cross-sectional study showed that exposure to increasing durations of DM was positively associated with sweet, fatty and salty taste preference in all participants, while inverse associations were observed for bitter TP, independently of diet quality and weight status. Differences by sex and age groups were observed.

### 4.1. Media Use in Association with Sweet Taste Preference

Our results showed that prolonged DM use was positively associated with high sweet TP in adolescents. These associations were also observed in the sensitivity analyses, where prolonged DM use in adolescents was associated with high sweet TP, regardless of their consumption frequency of sugary foods. This could be explained by the increase of media use with age [45] and, as a consequence, the higher exposure to food-related advertisements. Branding and TV marketing of high-sugar foods is associated with higher preference [21] and intake of those foods in children and adolescents [46]. Data from the same group of children included in our study, but at younger age (IDEFICS study- Identification and prevention of Dietary- and lifestyle-induced health EFfects In Children and infantS ), have shown that children with high TV and commercial exposures had a higher consumption of sugar sweetened beverages (SSB) [11,44] independently of parental norms. In our study, use of PC/game console was positively associated with females’ sweet TP, regardless of age. Similarly, in a longitudinal study conducted by Falbe et al. (2014), longer duration of electronic gaming in females was associated with increased frequency consumption of foods low in nutritional quality (e.g., sugar-rich foods), but not in males [47]. However, other individual differences might explain the female’s higher preference for sweet taste. Although no differences have been observed in the number of fungiform papillae between female and male children [48], it has been shown that females of older age can recognize taste intensity better than males, which could lead them to a heightened preference for sweet tasting foods [49].

### 4.2. Media Use in Association with Fatty Taste Preference

Positive associations were observed between exposure to DM (and the single media types) and fatty taste preference. High DM exposure in adolescents (especially females) was positively associated with high fatty TP. These associations remained in the sensitivity analyses, both in the high and low fat intake propensity groups, suggesting that DM use and exposure to its content could influence teens’ fatty TP, regardless of their actual intake of high-fat foods. Our results built on previous findings from earlier investigations when the IDEFICS participants were younger, which showed that high TV was associated with a higher propensity to consume fatty foods [9]. Children may also contribute to grocery shopping decisions (i.e. pester power), which in turn is associated with a high consumption of high-fat and high-sugar foods [50]. Moreover, those children who frequently asked for food/drink items seen on TV had a higher likelihood of later becoming overweight. Our results give evidence regarding a further hypothetical underlying pathway by which DM exposure could lead to poor eating habits and obesity, stressing the important role of taste preferences. This predisposition could be explained by neuropsychological factors, related to the sensory appeal of high-fat foods [51]. Studies have shown that unhealthy food cues (notably rich in fat content) attract children’s attention more than healthy ones [20]. Previous findings from the I.Family study showed that children watching unhealthy food images vs. healthy ones had increased activation in brain areas involved in reward, motivation and memory [52]. Literature suggests that personality traits related to urgency, lower levels of consciousness and higher levels of extraversion have been associated with preference for unhealthy foods [53] as well as with excessive screen time use in children [54]. Hence, it may be possible that children’s personality traits played a role in their preference for fatty foods. 

### 4.3. Media Use in Association with Bitter Taste Preference

Our results showed an inverse relationship between high DM use (TV, PC, smartphone and internet) and bitter taste preference. These findings, although cross-sectional, built on previous longitudinal studies indicating that extended screen viewing predicts lower intake of fruits and vegetables, with the latter being the responsible source for the bitter tasting molecules perceived by the taste receptors located on the tongue and other parts of the oropharynx [55]. TV food advertising was shown to lead to unhealthy dietary changes, including low intake of fruits and vegetables, which, despite their potential to promote health, receive little airtime [56]. In our study, prolonged exposure to PC, smartphone and internet was negatively associated with bitter TP, in adolescent males in particular, but not in females. One explanation could be that food marketing is more likely to influence males’ food preferences rather than that of females [57]. Furthermore, other factors related to family-environment might play a role in shaping children’s food preferences and eating patterns [58] as well as their screen time habits. Literature suggests that male children whose parents did not limit internet usage time, were at higher risk of developing internet addiction [59]. Another study based on I.Family participants observed that children with prolonged media use were more likely to come from non-traditional families with no rules set for screen time use [60]. On the other hand, parenting feeding practices and mother’s education can influence females’ eating habits [61], but not those of males [62]. Remarkably, other underlying social factors such as peer pressure and perception of body weight influence female adolescents to make healthier food choices compared to males [36].

### 4.4. Media Use in Association with Salty Taste Preference

Our study showed a positive trend in the association between high internet exposure and salty TP, especially in male adolescents. Studies have shown that male adolescents tend to play more advergames in a multiplayer gaming environment compared to younger males [63], hence being more exposed to digital advertising of HFSS food [64]. Coates et al. (2019) have shown that influencer marketing of unhealthy snacks in online social networking platforms is associated with increased intake of the promoted snacks [65]. Our results showed that female children who used PC for >2 h/day, cross-sectionally, had three times higher odds for preferring salty tasting foods compared to those using PC for ≤2 h daily. The broad confidence intervals suggest that these results should be interpreted cautiously. However, evidence has shown that female children are actually heavier users of PC games than female adolescents, hence they might also indulge more in snacking while gaming [66] including snacks with high salt content (e.g., potato chips and popcorn) [10].

### 4.5. Strengths and Limitations

This is the first epidemiological study evaluating associations between exposure to DM and its specific types in real-life setting and sensory taste preferences in European children and adolescents. We included information on TV, computer, game console, internet and smartphone use, thus having a broad picture of the media use patterns of the participants. One of the main strengths of our study is the large sample size of 7094 children and adolescents and the large age range (6 to 17 years) which enabled us to obtain reliable results. Including participants from seven European countries allowed us to have a clear understanding of the different types of media used across the continent and their potential influence on TP. As taste preferences were self-reported by adolescents, as well as by younger children (6 to 12 years), and not proxy-reported by parents, we could exclude parental social-desirability bias and recall bias in both age groups. Literature suggests that, when parents report food preferences for their children, they may report preferences similar to their own food preference [67]. The standardized protocol and the pre-test conducted in a subsample of children make the FBPQ an established and feasible instrument for evaluating preferences of food and drinks in children and adolescents [29]. Furthermore, using information on various covariates, such as country, sex, age, parental education status, migration background, diet quality and snack frequency consumption allowed us to adjust for potential confounders.

There are methodological limits to our investigation. We could not totally exclude a social-desirability bias as adolescents, who self-reported taste preferences tend to report less their liking of foods/beverages with high energy content, such as fat- and sugar-rich foods [68]. We could not obtain information on social media use and its specific platforms including Facebook, Instagram, Snapchat, TikTok and YouTube. The social networking sites are becoming ubiquitously present in children’s and adolescents’ everyday life and they represent a powerful gateway for food companies to advertise their unhealthy/junk products. Thus, we suggest that further research should tackle the influence of social media on children’s taste preferences. We did not distinguish between internet use for academic work and entertainment. This could explain the lack of significant association of internet use and sweet (and fat) taste preference in the overall sample. We acknowledge the limitation that mean media use in our children (2.4 h/day), is relatively low compared to current reports—U.S. children spend 5 h/day with screens while adolescents spend up to 7 h/day with recreational screen use [69]. However, as our data was collected during 2013–2014, the mean media use of our study is similar to that of earlier studies [70]. Newer studies with up-to-date information on media use in children and adolescents are warranted. Lastly, our research was conducted using cross-sectional data and we were unable to assess the temporal sequence in which dependent and independent factors occurred. Hence, future longitudinal studies with objectively-measured taste preferences are recommended to provide insights on the potential underlying mechanism by which exposure to DM content could influence poor eating habits and obesity in children and adolescents.

## 5. Conclusions

Exposure to DM was positively associated with increased preference for sweet, fatty, and salty taste while inversely associated with bitter TP in European children and adolescents. These results provide a starting point for future longitudinal research to shed light on further mechanisms by which exposure to DM might lead to poor eating behaviors and childhood obesity. Our findings could serve as an incentive for parents, pediatricians and policy makers alike in their battle to limit children’s and adolescents’ exposure to digital media content, to improve their eating habits and to prevent childhood obesity-related comorbidities.

## Figures and Tables

**Table 1 foods-10-00377-t001:** Characteristics of the study population by sex and exposure to digital media ^1^.

	Total Digital Media Exposure								All	
	≤2 h/day				>2 h/day					
	Sex									
	Males		Females		Males		Females			
	n	%	n	%	n	%	n	%	N	%
All	1401	19.7	1807	25.5	2101	29.6	1785	25.2	7094	100.0
Age group										
<12 Years	1017	14.3	1221	17.2	1009	14.2	765	10.8	4012	56.6
≥12 Years	384	5.4	586	8.3	1092	15.4	1020	14.4	3082	43.4
Parental educational status										
Low	90	1.3	82	1.2	107	1.5	98	1.4	377	5.3
Medium	569	8.0	718	10.1	947	13.3	792	11.2	3026	42.7
High	685	9.7	936	13.2	985	13.9	829	11.7	3435	48.4
Missing	57	0.8	71	1.0	62	0.9	66	0.9	256	3.6
Weight status										
Thin/Normal weight	1076	15.2	1370	19.3	1435	20.2	1250	17.6	5131	72.3
Overweight/Obese	320	4.5	435	6.1	661	9.3	533	7.5	1949	27.5
Missing	5	0.1	2	0.0	5	0.1	2	0.0	14	0.2
Migration background										
None of parents	1104	15.6	1396	19.7	1652	23.3	1396	19.7	5548	78.2
Both parents	77	1.1	110	1.6	109	1.5	99	1.4	395	5.6
One parent	134	1.9	175	2.5	192	2.7	165	2.3	666	9.4
Missing	86	1.2	126	1.8	148	2.1	125	1.8	485	6.8
HDAS										
High	701	9.9	908	12.8	1087	15.3	911	12.8	3607	50.8
Low	700	9.9	899	12.7	1014	14.3	874	12.3	3487	49.2
Snack frequency intake										
Low	782	11.0	1042	14.7	1014	14.3	889	12.5	3727	52.5
High	615	8.8	756	10.8	1073	15.3	879	12.6	3323	47.5
Sweet intake propensity										
High	648	9.1	770	10.9	1118	15.8	924	13.0	3460	48.8
Low	753	10.6	1037	14.6	983	13.9	861	12.1	3634	51.2
Fat intake propensity										
High	699	9.9	872	12.3	1093	15.4	820	11.6	3484	49.1
Low	702	9.9	935	13.2	1008	14.2	965	13.6	3610	50.9
Sweet TP										
Low	521	7.3	765	10.8	805	11.3	698	9.8	2789	39.3
High	879	12.4	1038	14.6	1293	18.2	1086	15.3	4296	60.6
Missing	1	0.0	4	0.1	3	0.0	1	0.0	9	0.1
Fatty TP										
Low	430	6.1	762	10.7	698	9.8	756	10.7	2646	37.3
High	970	13.7	1043	14.7	1402	19.8	1029	14.5	4444	62.6
Missing	1	0.0	2	0.0	1	0.0			4	0.1
Bitter TP										
Low	453	6.4	662	9.3	751	10.6	703	9.9	2569	36.2
High	933	13.2	1039	14.6	1329	18.7	955	13.5	4256	60.0
Missing	15	0.2	106	1.5	21	0.3	127	1.8	269	3.8
Salty TP										
Low	617	8.7	806	11.4	1050	14.8	849	12.0	3322	46.8
High	758	10.7	977	13.8	1013	14.3	918	12.9	3666	51.7
Missing	26	0.4	24	0.3	38	0.5	18	0.3	106	1.5
Country										
Italy	287	4.0	326	4.6	391	5.5	318	4.5	1322	18.6
Estonia	125	1.8	195	2.7	377	5.3	348	4.9	1045	14.7
Cyprus	272	3.8	344	4.8	486	6.9	461	6.5	1563	22.0
Sweden	106	1.5	161	2.3	238	3.4	174	2.5	679	9.6
Germany	240	3.4	315	4.4	272	3.8	212	3.0	1039	14.6
Hungary	231	3.3	280	3.9	244	3.4	209	2.9	964	13.6
Spain	140	2.0	186	2.6	93	1.3	63	0.9	482	6.8

^1^ HDAS-Healthy Dietary Adherence Score; TP—Taste preference.

**Table 2 foods-10-00377-t002:** Exposure to digital media in association with taste preferences in European children and adolescents ^1,2^.

	Sweet TP (N = 7085) ^3^		Fatty TP (N = 7090)		Bitter TP (N = 6825)		Salty TP (N = 6988)	
Media Types	Raw Model	Adjusted Model	Raw Model	Adjusted Model	Raw Model	Adjusted Model	Raw Model	Adjusted Model
	Odds Ratios (OR) and 95% Confidence Limits (95% CI)							
Total DM exposure (ref. ≤1 h/day)								
1–2 h/day	1.01 (0.87–1.19)	1.03 (0.88–1.21)	1.12 (0.96–1.31)	**1.19** **(1.01**–**1.41)**	0.82 (0.70–0.97)	**0.83** **(0.70**–**0.99)**	0.95 (0.81–1.11)	1.08 (0.92–1.27)
2–3 h/day	1.01 (0.86–1.18)	1.06 (0.89–1.25)	1.03 (0.87–1.21)	1.18 (0.99–1.40)	0.80 (0.67–0.94)	**0.81** **(0.67**–**0.96)**	0.78 (0.67–0.92)	1.00 (0.84–1.19)
>3 h/day	1.13 (0.97–1.33)	**1.23** **(1.03**–**1.46)**	1.11 (0.95–1.31)	**1.40** **(1.18**–**1.67)**	0.75 (0.64–0.89)	**0.72** **(0.60**–**0.87)**	0.82 (0.70–0.96)	1.15 (0.96–1.37)
TV viewing (ref. ≤1 h/day)								
1–2 h/day	1.02 (0.91–1.13)	1.02 (0.91–1.14)	0.98 (0.88–1.10)	1.00 (0.89–1.13)	**0.85** **(0.75**–**0.95)**	**0.86** **(0.77**–**0.97)**	0.95 (0.85–1.06)	1.01 (0.90–1.13)
2–3 h/day	1.24 (1.09–1.42)	**1.21** **(1.05**–**1.39)**	1.12 (0.98–1.28)	1.14 (0.99–1.31)	0.87 (0.76–1.00)	0.88 (0.77–1.02)	0.92 (0.80–1.04)	1.02 (0.88–1.16)
>3 h/day	1.20 (0.94–1.23)	1.20 (0.93–1.53)	1.08 (0.85–1.37)	1.17 (0.91–1.50)	**0.70** **(0.55**–**0.89)**	**0.74** **(0.58**–**0.95)**	1.01 (0.79–1.27)	1.19 (0.93–1.52)
PC use (ref. ≤1 h/day)								
1–2 h/day	0.93 (0.81–1.07)	0.96 (0.82–1.11)	0.96 (0.83–1.10)	1.01 (0.87–1.18)	0.94 (0.82–1.09)	0.89 (0.77–1.04)	0.79 (0.69–0.91)	0.91 (0.79–1.06)
2–3 h/day	0.98 (0.80–1.21)	1.04 (0.83–1.29)	1.04 (0.84–1.28)	1.17 (0.94–1.47)	0.84 (0.68–1.03)	**0.77** **(0.61**–**0.96)**	0.83 (0.68–1.02)	1.06 (0.85–1.32)
>3 h/day	1.04 (0.72–1.50)	1.15 (0.78–1.68)	1.37 (0.93–2.03)	**1.71** **(1.14**–**2.56)**	1.18 (0.80–1.72)	1.17 (0.78–1.74)	0.74 (0.52–1.06)	0.98 (0.67–1.43)
Smartphone use (ref. ≤1 h/day)								
1–2 h/day	0.82 (0.67–0.99)	0.91 (0.74–1.11)	0.79 (0.65–0.96)	0.90 (0.73–1.10)	0.91 (0.74–1.11)	0.84 (0.68–1.03)	0.72 (0.59–0.88)	0.80 (0.65–0.98)
2–3 h/day	1.06 (0.83–1.35)	1.16 (0.90–1.50)	1.17 (0.92–1.51)	**1.36** **(1.05**–**1.76)**	0.88 (0.68–1.13)	0.79 (0.60–1.03)	0.79 (0.62–1.01)	0.89 (0.69–1.15)
>3 h/day	0.95 (0.76–1.17)	1.10 (0.87–1.38)	1.02 (0.82–1.27)	**1.30** **(1.03**–**1.63)**	0.87 (0.69–1.08)	0.79 (0.62–1.05)	1.00 (0.81–1.24)	1.20 (0.96–1.51)
Internet exposure (ref. ≤1 h/day)								
1–2 h/day	0.88 (0.77–1.01)	0.93 (0.81–1.08)	0.79 (0.69–0.91)	0.94 (0.81–1.09)	0.90 (0.78–1.03)	**0.85** **(0.73**–**0.99)**	0.74 (0.64–0.84)	0.90 (0.78–1.04)
2–3 h/day	0.97 (0.82–1.15)	1.06 (0.88–1.27)	0.93 (0.78–1.10)	1.18 (0.98–1.41)	0.89 (0.75–1.06)	**0.80** **(0.66**–**0.97)**	0.85 (0.72–1.00)	1.12 (0.93–1.34)
>3 h/day	0.90 (0.68–0.95)	0.94 (0.78–1.14)	0.78 (0.66–0.93)	1.12 (0.92–1.35)	0.87 (0.72–1.03)	**0.80** **(0.65**–**0.97)**	0.81 (0.68–0.96)	1.13 (0.94–1.37)

^1^ Logistic regression models were adjusted for age group, sex, snack consumption, HDAS, parental educational status, migrant background and country, OR not reported. ^2^ TP—taste preference, DM—digital media, PC-computer/game console use. ^3^ N reported for single taste preferences in association with DM exposure. For the single media types, the N varied, due to the exclusion of participants not using that specific media type. Bold significance in the adjusted models is provided via confidence limits.

**Table 3 foods-10-00377-t003:** Exposure to digital media in association with sweet taste preferences, stratified by sex and age group ^1,2^.

Media Types	Adjusted Model			
	Males		Females	
	<12 Years	≥12 Years	<12 Years	≥12 Years
	Odds Ratios (OR) and 95% Confidence Limits (95% CI)			
Total DM exposure (ref. ≤2 h/day) ^3^ >2 h/day	0.91 (0.74–1.10)	1.25 (0.98–1.60)	1.06 (0.86–1.29)	**1.27** **(1.02**–**1.57)**
TV viewing (ref. ≤2 h/day) >2 h/day	1.12 (0.89–1.41)	1.20 (0.95–1.51)	**1.31** **(1.02**–**1.69)**	1.15 (0.91–1.45)
PC use (ref. ≤2 h/day) >2 h/day	0.89 (0.59–1.35)	1.07 (0.81–1.41)	1.21 (0.48–3.04)	1.48 (0.97–2.24)
Smartphone use (ref. ≤2 h/day) >2 h/day	2.52 (0.98–6.50)	1.27 (0.93–1.74)	1.43 (0.73–2.79)	1.00 (0.78–1.28)
Internet exposure (ref. ≤2 h/day) >2 h/day	1.03 (0.70–1.50)	1.07 (0.84–1.36)	0.99 (0.61–1.60)	1.02 (0.81–1.27)

^1^ Logistic regression models were adjusted for age (continuous), snack consumption, HDAS, parental educational status, migrant background and country, OR not reported. ^2^ DM-digital media, PC-computer/game console use. ^3^ 7085 participants included for total DM exposure (2023 male children, 1475 male adolescents, 1982 female children, 1605 female adolescents). For the single media types, the N varied, due to the exclusion of participants not using that specific media type at all (see Appendix A). Bold significance in the adjusted models is provided via confidence limits.

**Table 4 foods-10-00377-t004:** Exposure to digital media in association with fatty taste preference in European children and adolescents ^1,2^.

Media Types	Adjusted Model			
	Males		Females	
	<12 Years	≥12 Years	<12 Years	≥12 Years
	Odds Ratios (OR) and 95% Confidence Limits (95% CI)			
Total DM exposure (ref. ≤2 h/day) ^3^ >2 h/day	0.87 (0.71–1.06)	1.24 (0.96–1.61)	1.11 (0.90–1.36)	**1.37** **(1.10**–**1.70)**
TV viewing (ref. ≤2 h/day) >2 h/day	0.97 (0.77–1.23)	1.09 (0.86–1.39)	1.20 (0.93–1.54)	**1.28** **(1.02**–**1.61)**
PC use (ref. ≤2 h/day) >2 h/day	0.94 (0.62–1.42)	1.22 (0.92–1.62)	1.08 (0.45–2.60)	**1.83** **(1.21**–**2.76)**
Smartphone use (ref. ≤2 h/day) >2 h/day	1.52 (0.63–3.68)	**1.49** **(1.06**–**2.08)**	1.09 (0.57–2.08)	**1.36** **(1.07**–**1.75)**
Internet exposure (Ref. ≤2 h/day) >2 h/day	1.01 (0.68–1.48)	1.26 (0.98–1.61)	0.81 (0.51–1.29)	**1.37** **(1.10**–**1.71)**

^1^ Logistic regression models were adjusted for age (continuous), snack consumption, HDAS, parental educational status, migrant background and country, OR not reported. ^2^ DM-digital media, PC-computer/game console use. ^3^ 7090 participants included for total DM exposure (2024 male children, 1476 male adolescents, 1985 female children, 1605 female adolescents). For the single media types, the N varied, due to the exclusion of participants not using that specific media type at all (see Appendix A). Bold significance in the adjusted models is provided via confidence limits.

**Table 5 foods-10-00377-t005:** Association of media use with bitter taste preference in European children and adolescents ^1,2^.

Media Types	Adjusted Model			
	Males		Females	
	<12 Years	≥12 Years	<12 Years	≥12 Years
	Odds Ratios (OR) and 95% Confidence Limits (95%CI)			
Total DM exposure (ref. ≤2 h/day) ^3^ >2 h/day	0.82 (0.67–1.00)	0.79 (0.60–1.05)	1.07 (0.87–1.31)	0.82 (0.65–1.03)
TV viewing (ref. ≤2 h/day) >2 h/day	0.84 (0.68–1.06)	0.84 (0.65–1.07)	1.07 (0.84–1.38)	0.98 (0.77–1.25)
PC use (ref. ≤2 h/day) >2 h/day	1.31 (0.86–1.98)	**0.65** **(0.48**–**0.87**)	0.91 (0.36–2.32)	0.86 (0.56–1.31)
Smartphone use (ref. ≤2 h/day) >2 h/day	0.92 (0.42–2.03)	**0.68** **(0.49**–**0.94)**	0.80 (0.40–1.57)	0.98 (0.75–1.27)
Internet exposure (ref. ≤2 h/day) >2 h/day	0.94 (0.66–1.36)	**0.65** **(0.50**–**0.84)**	0.87 (0.53–1.43)	0.92 (0.73–1.16)

^1^ Logistic regression models were adjusted for age (continuous), snack consumption, HDAS, parental educational status, migrant background and country, OR not reported. ^2^ DM-digital media, PC-computer/game console use. ^3^ 6825 participants were included for total DM exposure (1995 male children, 1471 male adolescents, 1830 female children, 1529 female adolescents). For the single media types, the N varied, due to the exclusion of participants not using that specific media type at all (see Appendix A). Bold significance in the adjusted models is provided via confidence limits.

**Table 6 foods-10-00377-t006:** Association of media use with salty taste preference in European children and adolescents ^1,2^.

Media Types	Adjusted Model			
	Males		Females	
	<12 Years	≥12 Years	<12 Years	≥12 Years
	Odds Ratios (OR) and 95% Confidence Limits (95% CI)			
Total DM exposure (ref. ≤2 h/day) ^3^ >2 h/day	0.86 (0.71–1.04)	1.07 (0.84–1.38)	1.12 (0.92–1.37)	1.02 (0.82–1.27)
TV viewing (ref. ≤2 h/day) >2 h/day	0.92 (0.73–1.15)	1.05 (0.83–1.32)	1.16 (0.91–1.48)	1.06 (0.84–1.34)
PC use (ref. ≤2 h/day) >2 h/day	0.73 (0.49–1.10)	1.16 (0.88–1.53)	**3.85** **(1.26**–**11.72)**	1.14 (0.75–1.71)
Smartphone use (ref. ≤2 h/day) >2 h/day	0.99 ^3^(0.44–2.24)	1.21 (0.89–1.66)	1.62 (0.81–3.21)	1.00 (0.78–1.29)
Internet exposure (ref. ≤2 h/day) >2 h/day	1.14 (0.79–1.66)	**1.29** **(1.02**–**1.64)**	1.25 (0.77–2.02)	1.09 (0.87–1.36)

^1^ Logistic regression models were adjusted for age (continuous), snack consumption, HDAS, parental educational status, migrant background and country, OR not reported. ^2^ DM-digital media, PC-computer/game console use. ^3^ 6988 participants included for total DM exposure (1977 male children, 1461 male adolescents, 1953 female children, 1597 female adolescents). For the single media types, the N varied, due to the exclusion of participants not using that specific media type at all (see Appendix A). Bold significance in the adjusted models is provided via confidence limits.

## Data Availability

Data described in this study will be made available upon request from the corresponding author.

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
