# Peer review of "Digital Media Use in Association with Sensory Taste Preferences in European Children and Adolescents—Results from the I.Family Study"

_foods, 2021, doi:10.3390/foods10020377_

Round 1

Reviewer 1 Report

General comment

The manuscript investigates the association between children and adolescents exposure to digital media and food preferences for sweet, salty, fatty and bitter taste. It is a large scale study conducted in seven EU countries. The topic is interesting and worthy of study. The methods description permits other investigators to repeat the work. The statistical analysis of data is appropriate and the results are well presented and discussed. In my opinion, the overall scientific quality of the manuscript is good. 

Specific comments

L36-37. Please check the limits of the two age classes (12 years old subjects seem included in both groups).

L106. The study was conducted in 2013-2014 using. Is it correct? In L474, it is reported that data were collected during 2012-2013. Please check.

Table 2. Salty TP (6988) à Salty TP (N=6988)

L415-416 Bitter tasting molecules are not perceived by the olfactory system but by the taste receptors located on tongue. Please revise.

L474 “…was collected during 2012-2013”. Is that correct? In L106, it is reported that the study was conducted in 2013-2014. Please check.

L268-281 + Tab. 3. I would suggest to report the Body Max Index of the participants. From the I.Family project website, it seems that the BMI data were collected. After reading the introduction starting with the presentation of the relationship between the childhood obesity and the use of screen media devices, I would expect that the BMI was a variable included in the work. I wonder why the authors decided to not use that piece of information, since the BMI could be a relevant factor in influencing they results. Please provide information on BMI or explain why it is not taken into account.

Discussion. I appreciate that Authors try to explain the observed result focusing on motivations supporting their hypothesis. However, I would suggest to consider also potential other explanations that could not be related to the DM exposure but that could be due to different food preferences in general or to different individual factors, for instance the sex of the participants. E.g, it is known that females are more sensitive than males to food craving (Hormes et al. 2014, Appetite 83:185-193), to craving for sweet foods and they have a higher general interest for health (Monteleone et al. 2017, FQAP 59:123-140); for example, that could induce a higher general preference of females for sweet also when in front of a screen.

Strengthens.  The high number of subjects involved in the study allows to obtain very reliable results.

Limitations. The data presented are quite old, since they were collected in 2012-2013 (or 2013-2014?). Since the use of the DM is increased rapidly in the last years, I wonder if the reported data are still valid. Maybe more recent data collection could provide different results.

Author Response

Please see the attachment for the replies to Reviewer 1

Reviewer 2 Report

The manuscript “Digital media use in association with sensory taste preferences in European children and adolescents – results from the I.Family study” is a very interesting study presenting results about how digital use relates with taste preferences. This provides important information in a theme so actual, due to increasing exposure of children and adolescents to digital media.

The manuscript is well written and clearly presented. Authors did rigorous data analysis, which is clearly presented.

I only have some minor points that are essentially suggestions for authors, if they consider of interest.

Lines 62-64 - Please try to clarify this sentence, only by making slight changes. Suggestion: This behavior may promote adverse health effects in the current obesogenic environments with the omnipresence of HFSS foods together with a high exposure to digital environment.

Lines 70-71 - One recent study reported association between TV exposure, during meals, and lower preference for bitter and sour foods, suggesting a potential distraction of sensory aspects of meal when watching TV during eating. I remember this article that I read recently, when I was Reading this parto f the introduction, so think this could be an interesting reference to add here. (Rodrigues et al., 2020; https://doi.org/10.1080/09637486.2020.1738354)

Lines 70-71 - Probably obscuring not only the feeling of satiety but also the attention to sensory aspects, avoiding to became familiar with bitter and sour tastes and pair them to the positive post-ingestive signals after meals

Results – 1st paragraph - Authors refer Table 1 after the first sentence of the results section, but in fact Table 1 contains information about the main characteristics of the studies population and not only about age. I suggest to add a sentence, after the paragraph, stating that data about the characteristics of the population studied is presented in Table 1.

Lines 296-297 - I suggest to replace “in all participants” by “in both groups of participants”. I think this would improve clarity

Line 314 - I suggest “in both” instead of “in all”

Discussion - I would like to see referred in any part of the discussion that this positive association between sweet and fatty taste preference and DM duration can also have some relationship with psychological/personality traits of children. This has been not evaluated in this study, but we know that sweet and fatty foods are foods related with addiction. And addiction propensity is different among individuals. I consider that is not totally silly to consider that children with personality traits showing high tendency for become addicted to DM can also have higher tendency for other types of addiction, including food palatability.

Once more, I highlight that I am totally aware that this study did not evaluated personality parameters, but I consider of interest do not discard this possible additional explanation in discussion.

Lines 408-409 - This goes in line with my previous consideration that the effect of DM can differ according to personality traits and that the association between DM exposure and preference for palatable food can be a cause-effect relationship, but can also be the reflection of attraction for both types of “addictive behaviors” for the same individuals.

Author Response

Please see the attachment for the replies to Reviewer 2

Round 2

Reviewer 1 Report

The Authors improved the manuscript and considered all my suggestions. In my opinion, the current version of the manuscript is suitable for pubblication in Foods.